# Detecting Atypical Sentinel Lymph Nodes in Early-Stage Cervical Cancer Using a Standardized Technique with a Hybrid Tracer

**DOI:** 10.3390/cancers16152626

**Published:** 2024-07-23

**Authors:** Joana Amengual Vila, Anna Torrent Colomer, Catalina Sampol Bas, Adriana Quintero Duarte, Mario Ruiz Coll, Jorge Rioja Merlo, Octavi Cordoba

**Affiliations:** 1Gynecologic Oncology Unit, Obstetrics and Gynecology Department, Hospital Universitari Son Espases, 07120 Palma, Spain; atorrent@ssib.es (A.T.C.); mario.ruiz@ssib.es (M.R.C.); jorgeo.rioja@ssib.es (J.R.M.); octavi.cordoba@ssib.es (O.C.); 2School of Medicine, Universitat de les Illes Balears (UIB), 07120 Palma, Spain; 3Health Research Institute of the Balearic Islands, IdISBa, 07120 Palma, Spain; 4Department of Nuclear Medicine, Hospital Universitari Son Espases, 07120 Palma, Spain; catalinam.sampol@ssib.es; 5Department of Pathology, Hospital Universitari Son Espases, 07120 Palma, Spain; amquintero@ssib.es

**Keywords:** cervical cancer, sentinel lymph node, atypical lymphatic drainage, parametrial involvement, lymph node metastasis, hybrid tracer

## Abstract

**Simple Summary:**

The advantages of sentinel lymph node (SLN) biopsy over pelvic lymphadenectomy include reduced side effects like lymphoedema and lymphocyst and a more accurate detection of metastasis through ultrastaging. Another benefit includes the detection of alternative lymphatic drainage and SLN in atypical locations that would have been missed by standard pelvic lymphadenectomy. This is a prospective, observational, single-centre trial designed to determine the rate of atypical lymphatic drainage in patients with clinical early-stage cervical cancer (CC) using a hybrid tracer (ICG-^99m^Tc nanocolloid). The application of a hybrid tracer helps us to reach a high overall and bilateral SLN detection rate. The detection of nodes in unusual locations revealed a greater proportion of metastasis. Given the high percentage of metastases in atypical SLNs, without the SLN biopsy a significant proportion of patients would have been underdiagnosed and undertreated.

**Abstract:**

Background: Since October 2018, lymph node status has become part of the FIGO staging, given that it is one of the most important prognostic factors among women with CC. The aim was to determine the rate of atypical lymphatic drainage in patients with clinical early-stage cervical cancer using a hybrid tracer (ICG-^99m^Tc nanocolloid). Methodology: A prospective, observational, single-centre study conducted at Son Espases University Hospital between January 2019 and October 2023. Patients with clinical early-stage CC who underwent SLN mapping were included. External iliac and obturator nodes were defined as common locations. Para-aortic, common iliac, presacral, internal iliac, and parametrial nodes were defined as atypical locations. Results: Thirty-nine cases of CC were included. The overall SLN detection rate was 97.4%, with 89.5% bilaterally. Positive nodes were found in 21.1% of patients. Atypical lymphatic drainage was present in 8 out of 38 (21.1%) patients. Of all the SLNs biopsied (146), 10.3% corresponded to an atypical zone. SLN in the atypical area had a higher proportion of metastasis than the usual area (37.5% vs. 16.7%; *p* = 0.327). Conclusions: SLN biopsy can detect unusual drainage in a significant proportion of patients. Atypical lymph nodes have a higher percentage of metastasis, which consequently improves staging and tailoring therapy. SLN mapping performed via a standardized surgical technique using a hybrid tracer (ICG-^99m^Tc) could help in the identification of the “true SLN”.

## 1. Introduction

In October 2018, the International Federation of Gynaecology and Obstetrics introduced lymph node (LN) involvement in the cervical cancer (CC) staging. In the new classification, any patient with positive LN involvement automatically gets upstaged to stage IIIC. This new classification highlights LN involvement as one of the most important prognostic factors among women with early-stage CC. LN assessment allows tailoring treatment [1], selecting patients for surgery vs. chemoradiotherapy [2].

According to the 2023 National Comprehensive Cancer Network (NCCN) and ESGO/ESMO/ESP CC guidelines [3], the current standard treatment for early-stage CC (FIGO 2018 IA to IIA) consists of a radical hysterectomy (RH) and bilateral pelvic lymphadenectomy, with or without sentinel lymph node (SLN). SLN mapping has become a common and recommended practice during surgical staging of early-stage CC.

The SLN technique has confirmed that any pelvic LN can receive drainage from the cervical tumour [4]. Due to the midline location of the cervix in the pelvis, it has several drainage routes, with different authors providing a detailed description of the lymphatic pathways of the cervix and demonstrating that LN drainage is predictable but not uniform [5]. Eighty percent of resected SLNs are located in the external and obturator area, which are called common drainage SLNs since they are resected with pelvic lymphadenectomy. A non-negligible percentage of SLNs have been observed to be located in nodal chains that are not resected in systematic pelvic lymphadenectomy, such as those in the para-aortic, common iliac, parametrial, and presacral areas, which are called atypical drainage SLNs [6,7]. In the literature, articles describe up to 20% of SLNs as being located in atypical locations [8,9,10,11]. According to reports, 12–15% of recurrences occur when the complete pelvic lymphadenectomy performed during the original surgery is negative for metastasis [12,13]. Different explanations could be provided for these recurrences: (1) molecular techniques used to assess ultrastaging on all nodes are time consuming and expensive, thus limiting their routine use, (2) atypical lymph node metastases were not detected by standard lymphadenectomy and were left in situ, and (3) the presence of unfavourable tumour characteristics can be predictive of worse outcome (tumour size, deep stromal invasion, and lymph node involvement).

A few articles describe which factors are related to atypical drainage of SLNs [10,11]. One independent predictive factor for detecting SLNs in atypical location is the kind of tracer used [10]. The primary aim of this study was to determine the rate of atypical lymphatic drainage in patients with clinical early-stage CC using a hybrid tracer (ICG-^99m^Tc nanocolloid). Secondary outcomes included the following: estimating SLN detection rate, LN metastasis rate, empty LN packet, determining the number and location of SLNs, false-negative, sensitivity, accuracy, and negative and positive predictive value of the SLN technique.

## 2. Material and Methods

### 2.1. Study Group

A prospective, observational, descriptive, single-centre study was conducted at Son Espases University Hospital between January 2019 and October 2023. Patients with clinical early-stage CC who underwent SLN mapping during surgical staging were included. Physical gynaecological examination by an expert oncologic gynaecologist, pelvic MRI, and thorax-abdomen-pelvic CT or PET/CT scan were performed on all patients before surgery.

Selection and exclusion criteria are summarised in Table 1. All subjects provided their informed consent for inclusion before participating in this study. This study was conducted in accordance with the Declaration of Helsinki, and the protocol was approved by the Ethics Committee of the Balearic Islands (IB 4954/22 PI; IB 4955/22 PI).

### 2.2. Injection Tracer: ICG-^99m^Tc-Nanocolloid

SLN detection was performed with a hybrid tracer composed of 0.75 mL ICG added to a vial of ^99m^Tc-nanocolloid. The ICG-^99m^TC tracer was injected according to the same methodology described in a previously published article by our group [14].

### 2.3. Surgery

All women underwent laparoscopic (KarlStorz Endoscope, Tuttlingen, Germany) or robotic (da Vinci^®^ Surgical System Version Xi, Intuitive Surgical Inc., Sunnyvale, CA, USA) surgery for SLN procedures based on the SLN algorithm described in the NCCN guidelines [15,16]. A laparoscopic gamma probe compatible with a 10–12 mm diameter laparoscopic trocar was used to locate the radioactive intensity of ^99m^TC. Near-infrared fluorescence imaging in laparoscopic or Firefly mode provided by Da Vinci^®^ Surgical System was employed to identify ICG LN. During surgery, an SLN was defined when it was green and had increased activity detected via the gamma probe.

Pelvic drainage was examined after opening the entire retroperitoneal space, developing paravesical and pararectal spaces to expose the parametria, and identifying the external and internal iliac vessels, uterine artery, obturator nerve, and ureter. This approach allows a visual inspection, identifying visible fluorescence lymphatic channels from the cervix to the LN (ICG and hot SLN with an endoscopic gamma probe); SLN is defined as the first or second ICG-positive and ^99m^TC uptake node. It is always essential to look for and identify ICG-canaliculi to areas of atypical location—promontory area, common iliac artery, and paraaortic area—in order to avoid missing SLNs in atypical location. Reinjecting ICG was carried out if no SLN was finally identified. Pelvic lymphadenectomy was performed in cases where no tracer migration was detected.

The SLNs obtained were classified depending on the artery or vein they were adjacent to or according to anatomical area. External iliac and obturator nodes were defined as common SLN locations. Para-aortic, common iliac, presacral, internal iliac, and parametrial nodes were defined as atypical locations. Identified SLNs were removed selectively and sent for intraoperative histopathological and ultrastaging assessment by specialized gynaecologic pathologists.

Surgical treatment was modified based on the results of the intraoperative histological analysis of the SLN. In the event that the LN was found to be negative for metastasis, Querleu-Morrow RH type B/C was performed. If the LN was reported as positive, surgery on the uterus was abandoned and transperitoneal para-aortic lymphadenectomy was performed from the bifurcation of the aorta to the left renal vein, modifying the patient’s primary treatment to a concomitant chemo-radiation therapy protocol.

### 2.4. Histological Assessment

The intraoperative frozen section and ultrastaging protocol were performed for all SLNs. Subsequently, all SLNs were assessed pathologically according to the same methodology described in a previously published article by our group [14].

### 2.5. Data Collection and Statistical Analysis

Patients’ data were collected prospectively during this study by the main researcher (JAV). A data matrix was created by encrypting and integrating all information. Data statistical analysis was performed using IBM-SPSS v. 26 statistical software (IBM Corp., Armonk, NY, USA).

## 3. Results

### 3.1. Patients and Surgical Characteristics

Thirty-nine patients with clinical early-stage CC were included in the current study between January 2019 and October 2023. Thirty-eight patients had at least one SLN detected and were finally included for analysis (Figure 1). Patient and disease characteristics are presented in Table 2.

Squamous carcinoma was the most common histology (24, 63.1%), and more than half of patients had a tumour diameter ≤ 2 cm (24, 63.1%). On final pathologic examination, 10 patients (26.3%) had a tumour size larger than estimated with imaging or physical examination. After LN assessment and/or histological study of the radical hysterectomy (RH), 36.8% of patients were upstaged and 5.2% were downstaged, while 57.9% of patients were properly pre-surgically staged.

RH was performed on 32 patients (82%) due to the negative intraoperative results of the SLN and a trachelectomy on one patient (2.6%) with a history of previous subtotal hysterectomy and negative SLN. Para-aortic lymphadenectomy was carried out in six patients (15.4%) after positive intraoperative SLN; so, in these patients, RH was ruled out and they were treated with chemoradiotherapy.

### 3.2. Sentinel Lymph Node Assessment and Topography

Minimally invasive surgery (MIS) was used to perform the SLN technique. Robotic surgery was performed in 76.9% (30/39) of cases and laparoscopic surgery in 20.5% (8/39). Only one patient underwent laparotomy due to a concomitant colorectal carcinoma. In all instances, the hybrid tracer (ICG-^9^^9^^m^Tc-nanocolloid) was used. After LN evaluation assessment via MIS, RH was performed in 34 patients: 29.4% via laparoscopy and 67.6% using a “hybrid surgical approach”, within a research project, as explained in the article published by our group [14].

The overall SLN detection rate was 97.4%, with 87.2% of cases occurring bilaterally and 146 SLNs identified overall. The mean nodes per hemipelvis were 2.0 (1.0–3.0) on the right and 1.5 (1.0–2.0) on the left. SLNs were detected in more than one location in 13 (34.2%) patients. Positive SLNs were found in 21.1% (eight) patients. After the SLN biopsy, complete pelvic lymphadenectomy was performed in 22 (58%) patients; the reason was to validate the SLN technique at our centre. The median number of LNs obtained via lymphadenectomy was seven (range, 4–12). All the LNs excised during lymphadenectomy were negative in those patients, so there were no false-negative results; meanwhile, 87.5% of patients with metastatic SLNs had tumours larger than 2 cm. Two patients (6.7%; 2/30) were reported to have negative SLN intraoperatively, but after ultrastaging, micrometastases were found (size: 1 mm). Nodal size was similar between nodes with and without metastatic involvement, respectively [1.6 (1.0–1.8) vs. 1.5 (1.1–2.0)], *p* = 0.650. SLN results are described in Figure 1 and Table 3.

ICG was reinjected into five patients with migration failure (four unilateral and one with no migration). Drainage was obtained in two of them (2/5, 40%), and the obturator SLN was identified as being free of disease. Pelvic lymphadenectomy was carried out when there was no tracer migration detected and the entire resected LNs were negative. In one patient (2.6%) with ICG reinjection, green tissue was identified, but it did not yield a lymph node on pathological analysis (empty node packet (ENP)).

The distribution of SLNs was evaluated in relation to seven areas, and the results are summarised in Table 4. SLNs were mostly located in the external iliac area and obturator fossa, respectively [80 (54.8%); 51 (34.9%)]. Of the total SLNs (146), 10.3% corresponded to an atypical zone (the most frequent atypical area was the common iliac artery).

In short, atypical lymphatic drainage was present in 8 out of 38 (21.1%) patients (Table 4). The SLNs in the atypical area had a higher proportion of metastasis than the common area, but without reaching statistical significance (per patient: 37.5% vs. 16.7%, *p* = 0.327; per SLN: 20% vs. 4.6%, *p* = 0.051). Note that 75% exhibited macrometastases and 25% exhibited micrometastases detected via ultrastaging. All eight patients with atypical drainage were also associated with a common SLN. Most patients (7/8, 87.5%) had bilateral drainage whereas only one had unilateral drainage (this patient had two atypical drainages in one hemipelvis). Notably, two patients presented bilateral atypical drainage (in both hemipelvis).

Comparisons between patients with common and uncommon SLN locations are shown in Table 5. There was no significant difference in clinical characteristics or disease between the two groups. Patients’ characteristics with atypical SLNs are summarised in Table 6.

Parametrial nodes were detected in 17.9% (7/39) of women, four as SLNs and three in the histological study of the RH. One of the parametrial SLNs presented a macrometastasis. In this case, drainage was also observed to an SLN located in the same hemipelvis (external iliac) without the presence of metastases. In all cases where a parametrial SLN was detected, drainage was also observed to an SLN in a common location. There was no involvement of parametrial lymphatic channels or nodes in any patient (7/8, 87.5%) with a metastatic pelvic SLN.

After a follow-up period of 40.4 months (range 10–72 months) in July 2024, no recurrence was found.

## 4. Discussion

LN involvement is still the most significant prognostic factor for women with CC. However, as early-stage disease has a low risk of LN metastasis, in the last few decades, SLN mapping has gained more attention. Classically, the most commonly used tracer used to be ^9^^9^^m^Tc with or without blue dye. Nowadays, fluorescent mapping with ICG is significantly superior to ^9^^9^^m^Tc with blue dye or blue dye alone in terms of bilateral mapping in early-stage CC [18,19].

Our overall and bilateral SLN detection rate using a hybrid tracer is very high. It is important to note that our bilateral detection rate (87.2%) is higher than those larger studies, even though our results have to be interpreted with caution due to our study size (77%, 75%, and 62%, respectively) [17,20]. Our findings show that an SLN biopsy is highly sensitive (100%) in detecting metastases in early-stage CC. SLN had a negative predictive value (NPV) of 100%, and no false-negatives were observed in patients with pelvic lymphadenectomy after sentinel node mapping. This result could be explained by using a hybrid tracer (ICG-^9^^9^^m^Tc-Nanocolloid), the use of a standardized surgical technique, and strict involvement of both surgeons and Nuclear Medicine specialists.

Binding ICG to the radiocolloid (^9^^9^^m^Tc-nanocolloid-ICG) increases the time of its permanence in the SLN and decreases its migration to other non-SLNs, thereby improving and simplifying the technique. This method avoids the typical ICG lymphography and does not depend on the time between injection and sentinel node detection. The hybrid tracer combines the benefits of both modalities (fluorescence and radioisotope signal), enabling pre-surgical and intraoperative images to be obtained without additional side effects. By avoiding lymphography and using the double technique, we ensure the detection of the “true SLN”. With this hybrid tracer, we can perform pre-surgical SPECT/TC, which helps us locate drains and atypical SLNs. Consequently, it is possible for us to plan surgical approaches and time more effectively.

The ENP rate is not described in many articles. In our series, even though our study is limited by the small series, this rate (2.6%) is lower than the ones described by Frumovitz et al. [19] using ICG alone (5%) and Cabrera et al. [21] using ^9^^9^^m^Tc- ICG (4%) or ^9^^9^^m^Tc-methylene blue (0%). It has been reported that the percentage of ENP decreased with the increasing number of procedures performed. After approximately 30 procedures, the rate of empty packets seemed to have stabilized [22].

Despite the literature suggesting tumour size influences SLN detection rate—SLN detection rate was found to be statistically lower in patients with tumour sizes >3 cm than in patients with tumour sizes ≤ 3 cm (66.7% vs. 96.8%, *p* = 0.004) [9]—this was not the case in our study. In our series, detection was not associated with tumour size (86.9% ≤ 2 cm vs. 87.5% > 2 cm). In fact, neither age, BMI, FIGO stage, histology, tumour size, nor preoperative conization affected SLN detectability between common and uncommon locations.

In this study, in women with clinically early-stage CC, our LN positive rate was higher (21.1%) than the ones recently published by Jan Persson [23] (12.3%) and Pedro Ramirez (13.5%) [24], even though our study is limited by the small series. Articles in the literature describe atypical locations for up to 20% of SLNs [8,9]. The intraoperative anatomical localization of the SLNs in our review showed that SLNs were in a common location in 78.9% of cases, with the SLN most frequently located in the external iliac and obturator area. This distribution is in accordance with the published literature [6,8]. In contrast to Balaya’s 2019 report [11], which stated that there were no differences in the rate of positive SLNs among patients with or without SLNs in an atypical area, in our series, SLNs in the atypical area had a higher proportion of metastasis (37.5%) than the common area (16.7%). Despite the limitations of our small sample, our study is still important given that the SLN technique helps us to identify 21.1% of nodes in atypical locations, and, considering that a large proportion of them had metastasis, these patients were well-diagnosed and treated. Our findings support the benefits of the sentinel node technique in ensuring that patients with metastasis are detected and correctly treated.

It is important to mention the parametrium as a special group of atypical LNs. Note that the most frequent place where the parametrial sentinel node was located was above the uterine artery, just at its crossing with the ureter (Figure 2).

The lateral parametrium serves as the main pathway to the obturator and external iliac LN. The predicted lymphatic drainage pattern is a step-by-step progression from the cervical stroma to the LN located in the parametrium and posteriorly to the pelvic nodes, although there may be significant variation. Among atypical SLNs, the parametrial nodes are difficult to detect, with both vital dyes and/or radiotracer. The injection of ^9^^9^^m^Tc nanocolloid into the cervix, which is a point of high radioactivity, makes it challenging to detect isolated SLNs with a gamma probe. Studies show that ICG is a near-infrared fluorescence tracer with a high detection rate of parametrial sentinel nodes [25]. The importance of the detection of parametrial SLNs is under discussion, since the prognostic significance is not known, with some articles even denying that they have it, and excision of the parametrium is performed en bloc with the RH in order to achieve tumour-free margins [26]. At a time when de-escalation of surgical radicalism of CC surgery in the early-stages is under discussion, suggesting the possibility of conization or simple hysterectomy in tumours smaller than 2 cm, the detection of parametrial SLNs could gain importance [27,28].

The uncertain prognostic value of metastatic parametrial nodes without pelvic node involvement complicates the decision to categorize this situation as FIGO IIB or IIIC1; thereby emphasizing the requirement for a consensus on the most recent FIGO staging and presenting an alternative subgroup (IIIC1p: parametrial).

RH is based upon the notion that lymphatic channels and LNs are included in the parametrium. The tumour can directly involve parametrial tissue, or it can be indirectly involved through blood vessels or LN metastasis. The question to be resolved is whether juxtacervical nodes (parametrial tissue) in the pelvic lymphatic pathways should be considered true SLNs. It is still being debated whether parametrial nodes might be the first nodes to drain from primary cervical tumours [29].

Our review determined that parametrial LNs were detected as SLNs in only four patients (10.5%), which is higher than recent studies (3.9%, 0.5%, and 2%) [10,11,24], but lower than reported by others such as Lührs et al. (52%) [23]. In that case, the author explained the higher parametrial detection rate because of separate parametrial tissue removal and accurate assessment (ultrastaging and immunohistochemistry) of the parametrium. Our low detection rate of parametrial LNs could be explained with two reasons:(1)The real direct first site of lymphatic spread from the cervix is the pelvic nodes. There were no parametrial affected nodes found in any of the seven patients with positive pelvic sentinel nodes. Our study suggests that lymphatic drainage from the cervical tumour to the pelvic nodes could be a direct route that bypasses the parametrial nodes. Efforts have been made to determine whether a positive pelvic LN exists even when there are no positive parametrial nodes;(2)Parametrial SLNs are not easily identified with a gamma probe or ICG due to the high signal concentration of technetium or ICG in the cervix itself after injection of the tracer. The hybrid tracer could help with the identification of SLNs and ensure that we have identified the ‘true SLN’. The SLN was identified surgically by examining ICG and after removing the LN; once the node is outside the patient, the presence of technetium uptake could confirm that it is a parametrial SLN.

Despite the high percentage of parametrial nodes detected, only 2.1% and 3.1% of cases were reported as exhibiting metastasis at this level [23,29]. The value is not too far from our results (0.6%). However, if the parametria contain metastatic LNs, the nodes are removed and analysed en bloc at RH. Therefore, some authors advocate for parametrial resection separately and that its processed as SLN tissue. Finding parametrial node metastasis is significant since it indicates the need for adjuvant chemotherapy in patients who previously only received follow-up.

The main limiting factor of this study is the sample size. The strength of this series is its prospective nature, as we employ a standardized surgical technique and the same surgical team, which avoids biases as to what to consider an SLN.

Tracer methods are associated with finding more atypical SLNs. Since the ICG technique is new, it has only been primarily evaluated for this purpose by one study [10]. In future clinical trials, our group suggests that a hybrid tracer composed of ICG-^9^^9^^m^Tc-nanocolloid could be used to identify common and atypical lymphatic drainage, especially during the learning curve for choosing the true SLN, and to study the clinical importance of parametrial LNs.

## 5. Conclusions

The SLN technique has the advantage of detecting LNs in atypical locations that would have been missed with full pelvic lymphadenectomy. Due to a higher percentage of metastasis in atypical SLNs than in common ones, without the SLN technique, a larger percentage of women would have been underdiagnosed and undertreated. The SLN biopsy enabled us to stage the disease adequately and adapt the most appropriate treatment for each patient.

The SLN technique, performed via a standardized surgical technique using a hybrid tracer (^9^^9^^m^Tc-ICG), has a high detection rate for early-stage CC, excellent NPV, and high sensitivity. We suggest that SLN mapping with a hybrid tracer could help to detect true SLNs more accurately and decrease the ENP rate.

This study does not allow us to conclude whether removing the parametrium merely helps us to obtain a clear margin around the cervical tumour or whether it should be part of SLN assessment and, as such, be integrated in an SLN concept. Although more studies are needed, our series suggests that lymphatic drainage from the cervical tumour to the pelvic nodes could be a direct route that bypasses parametrial nodes.

## Figures and Tables

**Figure 1 cancers-16-02626-f001:**
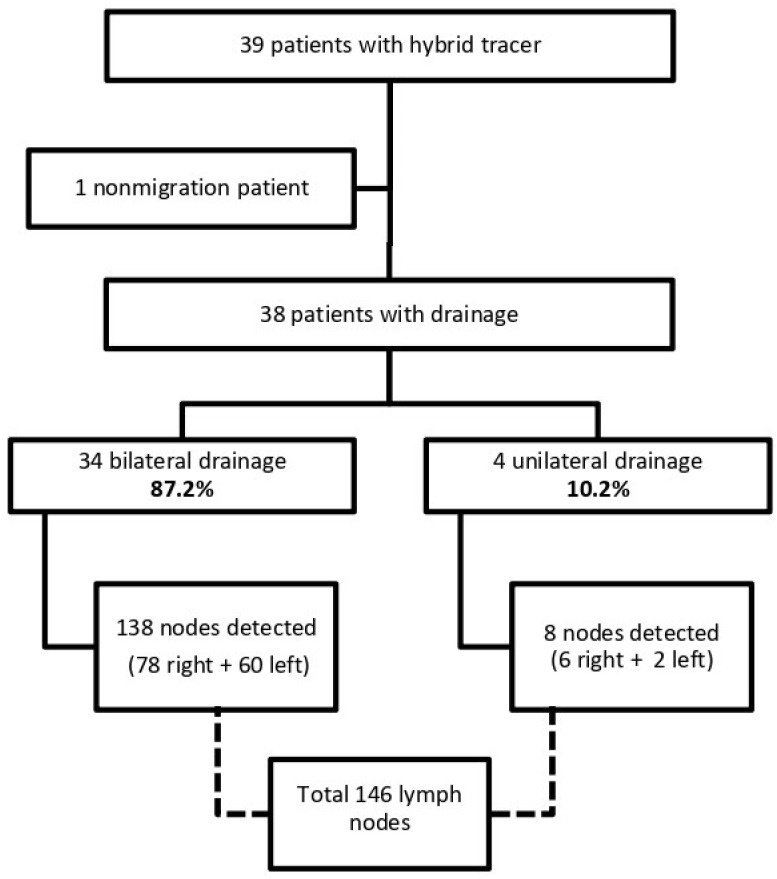
Flowchart showing patients included for analysis.

**Figure 2 cancers-16-02626-f002:**
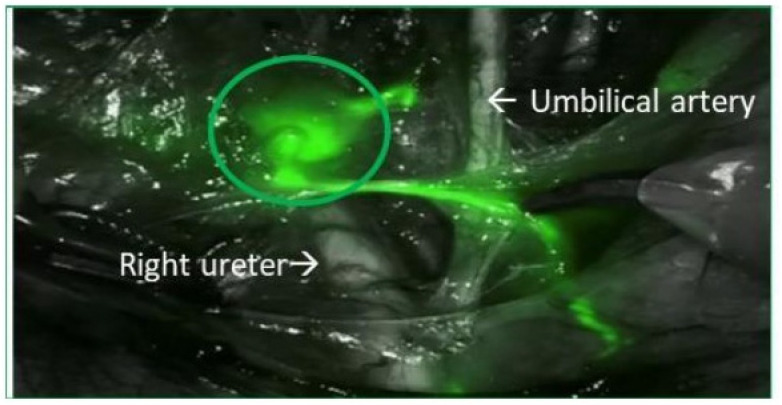
Parametrial sentinel lymph node.

**Table 1 cancers-16-02626-t001:** Selection and exclusion criteria.

SELECTION CRITERIA
(1) women over 18 years old
(2) histological diagnosis of adenocarcinoma, squamous carcinoma, and adenosquamous CC by biopsy or conization
(3) women with clinically early-stage CC (FIGO 2018 IA1 with LVI, up to stage IIA1)
(4) women who have undergone SLN mapping during surgical staging
(5) women having ECOG score performance status 0–2
(6) no contraindications for the surgical procedure
EXCLUSION CRITERIA
(1) locally advanced stages (FIGO IB3, II-IV; Except IIA1) at diagnosis
(2) metastatic disease
(3) medical co-morbidities contraindicated for surgical treatment
(4) history of pelvic radiotherapy
(5) pregnant women
(6) iodine hypersensitivity

CC = cervical cancer; FIGO = International Federation of Gynaecology and Obstetrics; LVI = lymphovascular invasion; SLN = sentinel lymph node; ECOG = Eastern Cooperative Oncology Group.

**Table 2 cancers-16-02626-t002:** Clinical Patient and Disease Characteristics.

Characteristics	Patients (N = 38)
Median age, yr. (range)	46.5 (40.0–54.0)
Body-mass index—Kg/m^2^	24.0 (22.0–27.0)
Conization	18/38 (47.4%)
Histology	
Squamous	24 (63.1%)
Adenocarc.	10 (26.3%)
Adenosquam.	4 (10.5%)
Preoperative tumor size	2.0 (0.5–3.0)
≤2 cm	24 (63.1%)
>2 cm	14 (36.8%)
FIGO 2018	
IA1/IA2	10 (26.3%)
IB1	14 (36.8%)
IB2	11 (28.9%)
IIA1	3 (7.9%)

Adenocarc. = adenocarcinoma; Adenosquam. = adenosquamous.

**Table 3 cancers-16-02626-t003:** Sentinel Lymph nodes’ detection rates.

	HYBRID (Tc99–ICG)
Total patient	39
Total included groins for SLN biopsy	146
SLN/patient	4 (3.0–4.0)
Right hemipelvis	2.0 (1.0–3.0)
Left hemipelvis	1.5 (1.0–2.0)
Overall pelvic detection	38/39 (97.4%)
Bilateral pelvic detection	34/39 (87.2%)
≤2 cm (≤IB1)	20/23 (86.9%)
>2 cm (≥IB2)	14/16 (87.5%)

Unilateral pelvic detection	4/39 (10.2%)
Not pelvic drainage	1/39 (2.5%)
Node involvement	0/5

SLN atypical drainage/patient	8/38 (21.1%)
SLN atypical drainage/node	15/146 (10.3%)
Empty node packet	1/38 (2.6%)
Positive SLNs/patient	8/38 (21.1%)
Positive typical SLN	5/30 (16.7%)
Positive atypical SLN	3/8 (37.5%)

False negative	0%
Sensitivity	8/8 (100%)
Accuracy	38/38 (100%)
Negative predictive value	100%
Positive predictive value	100%

Tc99 = 99mTc-nanocolloid; ICG = indocyanine green; SLN = sentinel lymph node.

**Table 4 cancers-16-02626-t004:** Topographic distribution and status of sentinel lymph nodes (n = 146).

Location	Intraoperative SLN Detection (n, %)	Metastasis (n, %)	Lécuru et al. J. Clin. Oncol. 2011 [17]
External iliac area	80 (54.8%)	5 (3.4%)	40%
Obturator area	51 (34.9%)	2 (1.36%)	40%
Internal iliac area	2 (1.4%)	0	
Common iliac	5 (3.4%)	0	10%
Presacral	3 (2.1%)	1 (0.6%)	5%
Parametrial	4 (2.7%)	1 (0.6%)	4%
Paraaortic	1 (0.7%)	1 (0.6%)	0.5%

SLN = sentinel lymph node.

**Table 5 cancers-16-02626-t005:** Comparison between patients with common and atypical nodes location.

Characteristics	Patients (N = 38)		
	Atypical Location (N = 8)	Typical Location (N = 30)	
Median age, yr. (range)	50.5 (40.5–63.0)	46.5 (39.0–54.0)	*p* = 0.410
Body-mass index—Kg/m^2^	23.5 (21.0–27.0)	25.5 (22.0–27.0)	*p* = 0.577
Conization	3/8 (16,8%)	15/30 (50%)	*p* = 0.697
Histology			
Squamous	7/8 (87.5%)	17/30 (56.6%)	
Adenocarc.	1/8 (12.5%)	9/30 (30%)	
Adenosquam.		4/30 (13.3%)	
Preoperative tumor size	2.0 (1.3–2.6)	2.0 (0.5–3.0)	*p* = 0.611
≤2 cm	5 (13.1%)	19 (50%)	
>2 cm	3 (7.9%)	11 (28.9%)	
FIGO 2018			NA
IA1/IA2	1 (10%)	9 (90%)	
IB1	4 (28.5%)	10 (71.4%)	
IB2	2 (18.1%)	9 (81.8%)	
IIA1	1 (33.3%)	2 (66.6%)	

Adenocarc. = adenocarcinoma; Adenosquam. = adenosquamous; NA = Not applicable.

**Table 6 cancers-16-02626-t006:** Patients’ characteristics with atypical sentinel lymph node (n = 8).

N	Age (Years)	BMI (Kg/m^2^)	Hystologic Type	Grade	Tumour Size (cm)	No. SLN	No. Positive SLN	Localization Atypical SLN	Positive Atypical SLN	Localization Common SLN	Positive Common SLN	Type Metastasis
1	62	24	Adenocarcinoma	G3	1.9	4	0	Common iliac	No	External iliac	No	X
2	64	23	Squamous	G3	2.5	3	1	Para-aortic; Common iliac	Yes	External iliac	No	Macro.
3	38	19	Squamous	G3	2.8	4	1	Presacral	Yes	External iliac	No	Macro.
4	36	50	Squamous	G3	1.2	4	0	Common iliac; presacral	No	Obturator	No	X
5	55	28	Squamous	G2	2.8	4	0	Parametrial; internal iliac	No	Obturator	No	X
6	43	20	Squamous	G1	0.25	4	0	Parametrial	No	External iliac; Obturator	No	X
7	46	22	Adenocarcinoma	G1	4	4	1	Parametrial	Yes	External iliac	No	Macro.
8	66	26	Squamous	G2	11	4	0	Parametrial	No	External iliac; Obturator	No	x

N = patient; BMI = body mass index; No. = number; SLN = sentinel lymph node; Macro = macrometástasis.

## Data Availability

The data that support the findings of this study are available on request from the corresponding author, [JAV]. The data are not publicly available due to restrictions apply to the availability of these data, which were used under license for this study.

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
