# Peer review of "Detecting Atypical Sentinel Lymph Nodes in Early-Stage Cervical Cancer Using a Standardized Technique with a Hybrid Tracer"

_cancers, 2024, doi:10.3390/cancers16152626_

Round 1

Reviewer 1 Report

Comments and Suggestions for Authors

Author Response

I am grateful that you have taken the time to review this manuscript. 

Reviewer 1 has not provided any comment or suggestion.  The PDF document has multiple lines or paragraphs that are highlighted in yellow. I'll attempt to clarify the relevant information:

  • Page 2: The location of references 6 and 7 has been modified due to their reference to the nomenclature of the typical and atypical location of sentinel nodes.

         I made corrections to reference 7, which was incorrect.

  • Page 2, : “is the kind of tracer used” is highlited in yellow. I'm uncertain about the suggestion.

         At this time, I want to resign that Kadan Y (reference 10) is the only author that found the type of tracer (ICG and a combination of Tc99 and Blue dye) as independent predictor factor for detecting sentinel lymph node in uncommon locations.

  • Page 4: It describes the composition of the molecule formed by two tracers together. The injection technique was already described by our group in an earlier article to which I refer.
  • Page 4: The entire section of surgery has been highlighted. I am confused about what clarifications I should make.
  • Page 5 and 6,. I am confused about what clarifications I should make.
  • Page 10, line 269-274. This paragraph outlines the advantages our group considers when using the hybrid tracer composed of Tc99 and ICG.

Additional clarificatios: 

If the reviewer considers that I have not answered any of the questions correctly, I apologize and, if he considers it appropriate, I undertake to answer them as soon as possible.

Reviewer 2 Report

Comments and Suggestions for Authors

The study is very well organised and provides strong evidence for the integration of SLN mapping into the standard surgical protocol for early-stage CC. The use of a hybrid tracer appears to improve detection rates and accuracy, which is critical for effective treatment planning. However, the limitations of the study, including the single-centre design and relatively small sample size, suggest that further multicenter studies with larger cohorts are needed to validate these results. Overall, this study makes an important contribution to the growing understanding of LN involvement in CC and the benefits of advanced SLN mapping techniques to improve patient prognosis and treatment precision.

The main purpose of the study is to determine the rate of atypical lymphatic drainage in patients with early-stage clinical cervical cancer using a hybrid tracer (ICG-99mTc-nanocolloid). The study highlights the use of a hybrid tracer combining indocyanine green (ICG) and technetium-99m (99mTc) nanocolloid for sentinel lymph node (SLN) mapping in early-stage cervical cancer. This method improves SLN detection rates, especially in the detection of atypical lymphatic drainage patterns. In addition, the study shows that a significant proportion of SLNs are located in atypical areas (21.1%) and that these nodules have a higher proportion of metastases than the usual areas. This finding emphasizes the importance of including atypical nodes in the investigation to improve staging and treatment. Prior to this study, there was limited information on the effectiveness of hybrid tracers in detecting SLNs in atypical sites. The research fills this gap by demonstrating that the hybrid tracer method can successfully identify these nodes, which have a higher likelihood of metastasis and are critical for accurate staging and tailored therapy in cervical cancer patients. By comparing their results with larger studies, the authors show that their bilateral detection rate is higher despite the smaller sample size. This comparison underscores the effectiveness of the hybrid tracer and the standardized technique used in this study. The study underlines the importance of a standardized surgical technique and close collaboration between surgeons and nuclear medicine specialists. This approach leads to a higher sensitivity (100%) and a higher negative predictive value (100%) for the detection of metastases, and thus sets a benchmark for future studies and clinical practice. A limitation of this study is the relatively small sample size, which may limit the generalizability of the findings. Future studies should aim to include a larger cohort of patients to enhance the statistical power and reliability of the results. However, the design of this study is very well organized: 1) inclusion of a control group that does not receiving the hybrid tracer (ICG-99mTc-nanocolloid) to compare the effectiveness and results of different SLN detection methods, 2) implementation of blinding procedures in which surgeons and pathologists are blinded to the type of tracer used, thereby reducing potential bias in the detection and evaluation of SLNs, 3) conducting the study at multiple centers to increase the diversity of the patient population and ensure that the results are more generally applicable. To improve further surveys, the authors could consider comparative analysis of different tracers (e.g. ICG alone, 99mTc alone and hybrid tracers) to assess their relative efficacy and safety. All references are appropriate. Regarding the tables and figures, I can suggest that the authors should be more attentive and create the tables with the same size and type of fonts so that they are more solid.

Author Response

I am grateful that you have taken the time to review this manuscript

Comment 1: tables and figures, I can suggest that the authors should be more attentive and create the tables with the same size and type of fonts so that they are more solid.

Response1:

I would like to apologize for this mistake that was not caught. At present, table and figure titles and contents are written in the same type and font size as the rest of the text (Times New Roman 10).

A new PDF with the modifications will be attached.

Comment 2: A limitation of this study is the relatively small sample size, which may limit the generalizability of the findings. Future studies should aim to include a larger cohort of patients to enhance the statistical power and reliability of the results. However, the design of this study is very well organized:

1) inclusion of a control group that does not receiving the hybrid tracer (ICG-99mTc-nanocolloid) to compare the effectiveness and results of different SLN detection methods,

2) implementation of blinding procedures in which surgeons and pathologists are blinded to the type of tracer used, thereby reducing potential bias in the detection and evaluation of SLNs,

3) conducting the study at multiple centers to increase the diversity of the patient population and ensure that the results are more generally applicable. To improve further surveys, the authors could consider comparative analysis of different tracers (e.g. ICG alone, 99mTc alone and hybrid tracers) to assess their relative efficacy and safety.

Response 2:

The aim of this Project was to determine the rate of atypical lymphatic drainage in patients with clinical early-stage cervical cancer using a hybrid tracer (ICG-99mTc nanocolloid).

This is a single-center study with a limitation, but conclusive results were obtained based on the size of the analyzed sample. This weakness is identified and observed in the discussion.

Small sample limits the achievement of better evidence. The results obtained serve as a basis for proposing new studies of greater sample that allow rescue of the minimally invasive pathway in cervical cancer and the use of a hybrid tracer.

Based on this study, we conclude that randomized prospective studies are necessary to assess the our results. So, our group has designed a   prospective, multicenter, randomized Study (collaboration with Hospital Clínic de Barcelona) to compare SLN mapping using only ICG as a tracer vs. hybrid tracer (99m Tc-nanocolode-ICG). The surgical team will not be able to access the SPECT-CT when hybrid tracer is used.

Reviewer 3 Report

Comments and Suggestions for Authors

In this interesting manuscript the authors describe the amount of atypical sentinel nodes using a hybrid tracer technique in early cervical cancer. They found atypical pathways in 21% and statistically no difference in positive SLN detection depending on the pathway.

There is one major and some minor flaws in this manuscript that should be adressed.

Major flaw:

Because of the relatively high % of atypical flow with the hybrid technique the authors state that, by not using this technique, patients would be undertreated, resulted in worsened prognosis (page 10, lines 255/256). This statement is too bold, because when a SLN procedure is not carried out and a radical hysterectomy with full pelvic lymphadenectomy would have been carried out, common iliac nodes are always removed as well as parametrial nodes, and most likely also the nodes in the internal iliac area. Only presacral nodes and pao nodes will not be removed in a standard radical hysterectomy procedure. If this will result in a poorer outcome is very unlikely. I would advise the authors to be more cautious and downgrade this statement.

Minor:

page 2, lines 67/68. Are the references (8,9) as used for this statement, the right ones? Reference 8 does not mention the rate of recurrences in patients without apparent LN involvement. Reference 9 describes 68 patients who had a sentinel lymph node procedure before trachelectomy. Two recurrences occurred in the group who had a negative SLN. They did not undergo a full pelvic lymphadenectomy. Please explain why you use these references?

page 3, Table 1 and 2. Table1 tells us that stage IB3 will be excluded, but in Table 2 there are 2 patients with stage IB3. Please explain.

page 10, line 218: Please explain how you get to the false negative rate of 0% while, as far as I could figure out from the methods section, you did not perform a full pelvic lymphadenectomy in case the SLN was negative? There is also no follow-up available, consequently there is no info available on (pelvic) recurrences?

Author Response

I am grateful that you have taken the time to review this manuscript. First, I have added new references to the introduction as suggested. Secondly, I will make an effort to clarify and adjust the major and minor suggestions that were mentioned.

Comment 1:

Because of the relatively high % of atypical flow with the hybrid technique the authors state that, by not using this technique, patients would be undertreated, resulted in worsened prognosis (page 10, lines 255/256). This statement is too bold, because when a SLN procedure is not carried out and a radical hysterectomy with full pelvic lymphadenectomy would have been carried out, common iliac nodes are always removed as well as parametrial nodes, and most likely also the nodes in the internal iliac area. Only presacral nodes and pao nodes will not be removed in a standard radical hysterectomy procedure. If this will result in a poorer outcome is very unlikely. I would advise the authors to be more cautious and downgrade this statement.

Response 1:

I am in agreement with the comment that the tone of this statement is too harsh. Taking into account the limitations of the sample study, I have made changes to this sentence.  In the modified manuscript I have changed lines (page 10):

Despite the limitations of our small sample, our study is still important given that thanks to SLN technique helps us to identify 21.1% of nodes in atypical locations; and, considering that a large proportion of them had metastasis, these patients were well-diagnosed and treated.

This is a single-center study with a major limitation: small sample. But conclusive results were obtained based on the size of the analyzed sample. This weakness is identified and commented in the discussion. Small sample limits the achievement of better evidence.  The results obtained serve as a basis for proposing new studies. Based on this study, we conclude that randomized prospective studies are necessary to assess our results.

One of the main benefits of SLN mapping is identify the first lymph node that can be affected by metastasis and using ultrastaging protocols improve detection of micrometastasis that is not been posible when full pelvic lymphadenectomy have been carried out. SLN mapping reduce the extent of surgery and the associated complications of systematic pelvic lymphadenectomy.

According to the literature there are controversies regarding the limits of pelvic lymphadenectomy. The proximal limit has been variably described between the bifurcation of the common iliac to its origin at the bifurcation of the aorta. Some authors include level II corresponding to the common iliaca (1), but other authors define the upper limit the bifurcation of the common iliac artery(2).

Sacral lymph nodes are generally not encountered in the pelvic lymph node group and its dissection is not a routine part of pelvic lymphadenectomy.

  • Cibula D, Abu-Rustum NR. Pelvic lymphadenectomy in cervical cancer-surgical anatomy and proposal for a new classification system. Gynecol Oncol. 2010;116:33.
  • Sideri M. Surgery for Cervical Neoplasia. Morrow’s gynecologic cancer surgery. 2nd ed. South Coast Medical Publishing. 2012.

Parametrial nodes: I  mentiones the parametrium as a special group of atypical LNs. In the discussion I raise the debate of whether the parametrium should be part of the evaluation of the nodal status (sentinel node vs lymphadenectomy) or part of the radical hysterectomy. We have detected four paramentrial SLN and just one have metastasis. It is true that it is only one case and that when performing the radical hysterectomy we would have identified it. But this patient, thanks to the detection of this metastasis, was a candidate to abandon the radical hysterectomy and perform preaortic lymphadenectomy. If the sentinel node had not been performed, we would have had a combination of treatments (surgery and chemoradiation therapy) that is not recommended. Drawing any conclusions is impossible due to the small sample. Therefore, I am leaving it up to future studies and discussion.

Comment 2: page 2, lines 67/68. Are the references (8,9) as used for this statement, the right ones? Reference 8 does not mention the rate of recurrences in patients without apparent LN involvement. Reference 9 describes 68 patients who had a sentinel lymph node procedure before trachelectomy. Two recurrences occurred in the group who had a negative SLN. They did not undergo a full pelvic lymphadenectomy. Please explain why you use these references?

Response 2:

Thank you for pointing this out. Reviewing this paragraph I have noticed my error in the placement of references and in their explanation. The placement of references 8 and 9 is incorrect. The percentage of lymph nodes in uncommon locations is supported by both references and others that I included (8-11).

In the literature, articles describe up to 18-20% of SLNs in atypical locations8-11.

As for the explanation of 15% of recurrences in patients with early-stage cervical cancer, I added the correct reference and made the corresponding explanations that could justify recurrence in this type of patient.

According to reports, 12-15% of recurrences occur when the complete pelvic lymphadenectomy performed during the original surgery is negative for metastasis12 13. Different explanations could be given for these recurrences: (1) molecular techniques used to assess ultrastaging on all nodes are time consuming and expensive thus limiting its routine use, (2) atypical lymph node metastases were not detected by standard lymphadenectomy and were left in situ, (3) the presence of unfavorable tumor characteristics can be predictive of worse outcome (tumor size, deep stromal invasion, and lymph node involvement).

Comment 3: page 3, Table 1 and 2. Table1 tells us that stage IB3 will be excluded, but in Table 2 there are 2 patients with stage IB3. Please explain.

Response 3:

Thank you for bringing this to my attention. Checking the database, I detected this error. There was a mismatch between physical and radiological examination (MRI informed at 3.7 and 3.5 cm tumor, FIGO stage IB2) while physical examination were recorded 4 cm and classified IB3, this last data was reflected in the database. Postoperative study of radical hysterectomy both was tumors less than 4 cm so they are considered initial. I have classified them (by radiological size as IB2).

Table II and IV are modified.

The tables in the new manuscript have not been modified, but I have included a new file that contains the corrected ones.

Comment 4

page 10, line 218: Please explain how you get to the false negative rate of 0% while, as far as I could figure out from the methods section, you did not perform a full pelvic lymphadenectomy in case the SLN was negative?

Response 4:

Between the lines 216-219 ( SLN assessment subheading of the results) can be found the explanation: after the SLN biopsy, complete pelvic lymphadenectomy was performed in 22 (58%) patients; the reason was to validate the SLN technique at our centre. The median number of LNs obtained by lymphadenectomy was seven (range, 4-12). All the LN excised during lymphadenectomy were negative in those patients, so there were no false-negative results.

Page 10 is modified: no false negative rate were observed in patients with pelvic lymphadenectomy after sentinel node mapping.

Comment 5: There is also no follow-up available, consequently there is no info available on (pelvic) recurrences?

Response 5:

The aim of this study was to determine the rate of atypical localizations of sentinel lymph node in patients with early-stage cervical cancer. This is why the period of inclusion has been between January 2019 and October 2023. To assess the clinical impact, we need almost 2-3 years of follow-up.

After a follow-up average of 40.4 months (range 10-72 months), no recurrence was detected at the time of publication of the article (if accepted).

Page 8: After a follow-up period of 40.4 months (range 10-72 months) in July 2024, no recurrence was found.

Round 2

Reviewer 3 Report

Comments and Suggestions for Authors

No further comments, the authors responded adequately to the reviewers comments.